analytical chemistry/computational chemistry/ green chemistry

terbinafine, itraconazole, synchronous fluorescence spectroscopy, spiked human plasma

**Author for correspondence:**
Heba Elmansi
e-mail: dr_heba85@hotmail.com

# Combining derivative and synchronous approaches for simultaneous spectrofluorimetric determination of terbinafine and itraconazole

Heba Elmansi, Aya Roshdy, Shereen Shalan and Amina El-Brashy

Department of Pharmaceutical Analytical Chemistry, Faculty of Pharmacy, Mansoura University, 35516 Mansoura, Egypt

HE, 0000-0002-3953-7169; SS, 0000-0002-1468-1367

In this study, determination of terbinafine and itraconazole down to biological concentration level has been carried out. The determination is based on increasing the selectivity of the spectrofluorimetric technique by combining both derivative and synchronous spectrofluorometric approaches, which permits successful estimation of terbinafine at 257 nm and itraconazole at 319 nm in the presence of each other at $\Delta\lambda$ of 60 nm. International Conference on Harmonization validation guidelines were followed to fully validate the method, and linearity was obtained for the two drugs over the range of 0.1–0.7 µg ml$^{-1}$ for terbinafine and 0.5–4.0 µg ml$^{-1}$ for itraconazole. Application of the method was successfully carried out in the commercial tablets with good agreement with the comparison spectrofluorometric methods. As the detection limits were down to 0.013 and 0.1 µg ml$^{-1}$ and quantitation limits were 0.04 and 0.032 µg ml$^{-1}$ for terbinafine and itraconazole, respectively; the *in vitro* determination of terbinafine and itraconazole in spiked plasma samples was applicable. The percentage recoveries in biological samples were 97.17 ± 4.54 and 98.75 ± 2.25 for terbinafine and itraconazole, respectively. Water was used as the optimum diluting solvent in the proposed methodology which adds an eco-friendly merit.

# 1. Introduction

The sensitivity is an important merit in spectrofluorometric determination; however, overlapping excitation and emission spectra is a common problem. Therefore, synchronous fluorescence spectroscopy (SFS) gained great interest because of its advantages, e.g. enhancement in selectivity, simplicity of measuring and improvement in spectral resolution. It depends on the simultaneous scanning of the emission and excitation fluorescence spectra [1]. Herein, SFS is combined with derivative spectroscopy to determine the mixture of terbinafine and itraconazole.

Terbinafine HCl (TRH) is (2E)-N,6,6-Trimethyl-N-(naphthalen-1-ylmethyl) hept-2-en-4-yn-1-amine hydrochloride, figure 1a. It is freely soluble in methylene chloride and methanol [2]. Different methods were reported in the literature for its determination, including spectrophotometric [3,4], spectrofluorimetric [5] and chromatographic methods [6–9].

Itraconazole (ITR) is 4-[4-[4-[4-[[cis-2-(2,4-dichlorophenyl)-2-(1H-1,2,4-triazol-1-ylmethyl)-1,3-dioxolan-4-yl] methoxy] phenyl] piperazin-1-yl] phenyl]-2-[(1RS)-1-methylpropyl]-2,4-dihydro-3H-1,2,4-triazol-3-one, figure 1b. It is freely soluble in methylene chloride and sparingly soluble in tetrahydrofuran [2]. Itraconazole was determined by spectroscopic methods [10–13]. Different chromatographic methods were used for determination of itraconazole including GC/MS-SIM, LCMS/MS and HPLC [14–19].

Terbinafine and itraconazole are commonly used for treatment and prophylaxis of systemic fungal infection. These agents act by inhibiting ergosterol synthesis that has an important role in the synthesis pathway of fungal cell wall [20], so this leads to cell death by inhibiting fungal and bacterial cell wall. Some studies prove the synergistic effect of this combined therapy in treatment of chromoblastomycosis caused by melanized fungi [21–23].

The concurrent determination of TRH and ITR has been discussed in two reports [24,25]. The two reported methods are based on using acetonitrile as a main solvent. Additionally, the reported HPLC method [25] has a long run time (12 min). Our aim was to develop a new sensitive and smart methodology for their simultaneous quantitation. The present study is suggesting a new spectrofluorometric method for TRH and ITR determination for the first time using first derivative SFS. It is a successful way to quantitate them with satisfied accuracy and precision in tablets and biological fluids. Additionally, the method is facile and cost-effective as it uses a technique available in most laboratories. Water is the main diluting solvent in the proposed methodology which adds an eco-friendly impact.

# 2. Experimental procedure

## 2.1. Apparatus, materials, solvents and reagents

— For spectrofluorimetric measuring, Cary Eclipse fluorescence spectrophotometer equipped with xenon flash lamp was used. The voltage was 800 V and slit width was 5 nm. All obtained spectra were smoothed with factor = 20. Cary Eclipse software from Agilent Technologies was used for manipulation of data.
— pH meter (Consort, Belgium) was used.
— Vortex mixer (IVM-300p, Taiwan) and centrifuge (2–16P, Germany) were used for biological samples.
— Terbinafine HCl was provided by Novartis Pharma AG, Basle, Switzerland.
— Itraconazole was obtained from Multi Apex Pharmaceutical Industries-S.A.E, Badr City, Egypt.
— Lamisil® 125 mg tablet (Novartis Pharma S.A.E Cairo-C.C.R.111108), containing 125.0 mg of TRH as labelled.
— Itranox® capsules, labelled to contain 100.0 mg ITR in each capsule, product of Multi Apex Pharmaceutical Industries-S.A.E, Badr City, Egypt.
— Human plasma samples were kindly provided by Mansoura University Hospitals (Mansoura, Egypt) and kept frozen. The samples were subjected to gentle thawing before use.
— Syringe filters (Minisart RC25)—0.45 µm pore size were purchased from Sartorius-Stedim (Göttingen, Germany).
— Methanol, acetonitrile and n-propanol were purchased from Sigma-Aldrich (Germany).
— Surfactants as sodium dodecyl sulfate (SDS), cetrimide, and Tween 80 and chemicals used for buffer preparations were bought from El Nasr chemical Co., Egypt. Acetate and borate buffers were prepared at concentration of 0.2 M for each.

(a)

(b)

**Figure 1.** Structural formulae of studied drugs: (a) TRH and (b) ITR.

## 2.2. Standard solutions

In order to prepare stock solutions with concentration of 100.0 µg ml$^{-1}$, 10.0 mg of each of TRH and ITR were dissolved separately in 100 ml methanol in volumetric flasks. Subsequent dilution with methanol was carried out to get the working solutions.

## 2.3. Procedures

### 2.3.1. Construction of calibration graphs

Aliquots from the working solutions of TRH and ITR were transferred into a series of 10 ml volumetric flasks. The solutions were then completed with distilled water to volume to reach the ranges of 0.1–0.7 µg ml$^{-1}$ and 0.5–4.0 µg ml$^{-1}$ for TRH and ITR, respectively. Synchronous measurements were carried out at $\Delta\lambda = 60$ with scanning range 200–500 nm. The first-order derivative spectra ($^1D$) were operated using filter size 20.0 and 1.0 nm interval. The amplitudes were estimated at 257 and 319 nm for TRH and ITR, respectively, considering the blank reading. In order to construct the calibration graphs; ($^1D$) amplitudes were plotted against the final drug concentration in µg ml$^{-1}$ and regression analysis was carried out.

### 2.3.2. Analysis of TRH and ITR synthetic mixtures and pharmaceutical preparations

Aliquots with varied ratios of TRH and ITR were analysed as described under 'Construction of calibration graphs'. Then, the percentage recoveries were calculated by referring to the calibration curves or the regression equations.

For tablet assay, 10 Lamisil® or Itranox® tablets were subjected to weighing, mixing and grinding. Ten milligrams TRH or ITR were transferred into a 100 ml volumetric flask, 70 ml methanol was added and sonicated for 30 min. The flasks were completed to volume and filtered to be assayed as illustrated in 'Construction of calibration graphs'. ($^1D$) were calculated, and the content of tablets was computed by the regression equations.

### 2.3.3. Analysis of TRH and ITR in spiked human plasma

Simultaneous determination of TRH and ITR in spiked human plasma was carried out referring to their therapeutic levels [26]. One millilitre plasma was transferred separately into a set of

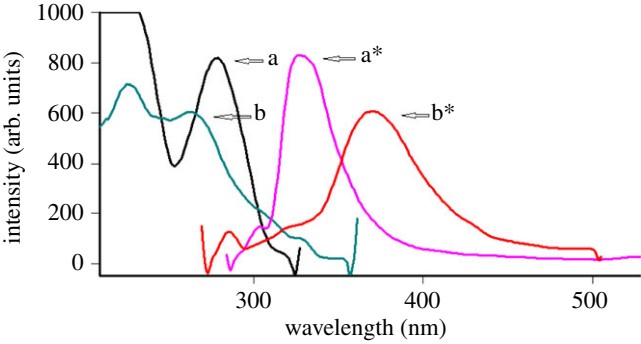

**Figure 2.** Excitation and emission fluorescence spectra of two drugs: (a–a*) TRH and (b–b*) ITR.

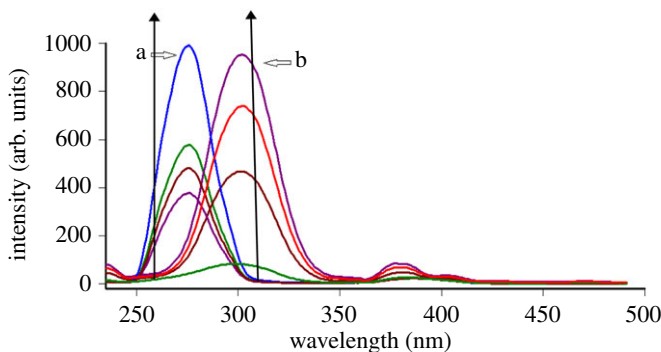

**Figure 3.** Synchronous fluorescence spectra of: (a) TRH (0.3, 0.35, 0.45 and 0.7 µg ml⁻¹) and (b) ITR (0.7, 1.7, 2.7 and 4.0 µg ml⁻¹).

centrifugation tubes. Aliquots from TRH and ITR stock solutions were added to reach final concentrations 0.3–0.7 and 0.7–1.0 µg ml⁻¹ for both drugs, respectively. Four millilitres methanol were added and then the tubes were subjected to vortex mixing for 5 min and centrifuged for another 30 min at 4000 r.p.m. to allow complete separation of the drugs from plasma contents. The upper clear layer was filtered through 0.45 µm syringe filters. One millilitre aliquots from the filtrate were quantitatively moved into a set of 10 ml volumetric flasks and diluted with distilled water to the volume. Measurement of the drug concentration was carried out as the procedure defined under 'Construction of calibration graphs' with blank sample in parallel. The amplitudes were plotted versus the concentration of each drug in µg ml⁻¹.

# 3. Results and discussion

Based on the importance of synchronous fluorimetry in selectivity and resolution enhancement, we aimed to use this approach to quantify the commonly co-administered drugs TRH and ITR simultaneously in their different matrices. Better performance was observed when combining both synchronous fluorimetry and derivative approaches. Figure 2 illustrates the overlapping spectra of TRH and ITR, where TRH excitation and emission wavelengths are 275 and 336 nm, respectively, and ITR has an emission band at 380 nm when excited at 260 nm.

The synchronous fluorescence spectra of increasing concentrations of TRH and ITR show some overlapping that does not permit their accurate measuring, specially for TRH. As is noted from figure 3, the point at which TRH could be measured without interference from ITR is not at a peak maximum. This leads us to try the first derivative SFS as a facile approach for enhancing spectral resolution and selectivity so the fluorescence spectra of TRH and ITR could be well separated. TRH could be measured quantitatively at 257 nm in the presence of ITR (figure 4), and ITR at 319 nm in the presence of TRH (figure 5), under the described experimental conditions.

## 3.1. Optimization of experimental conditions

For providing better performance, different variables affecting the spectra of TRH and ITR were investigated as given below.

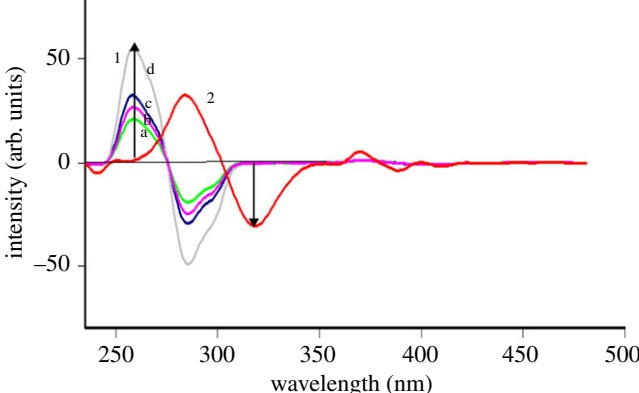

**Figure 4.** First derivative synchronous fluorescence spectra of: (1) (a–d) of TRH (0.3, 0.35,0.45 and 0.7 µg ml$^{-1}$) at 257 nm and (2) ITR (2.7 µg ml$^{-1}$).

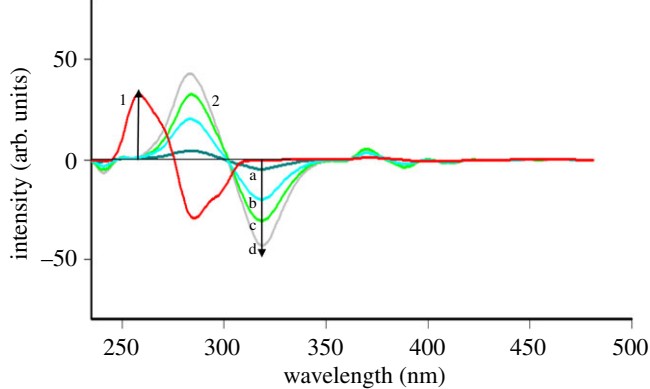

**Figure 5.** First derivative synchronous fluorescence spectra of: (1) TRH (0.45 µg ml$^{-1}$) and (2) (a–d) of ITR (0.7, 1.7, 2.7 and 4.0 µg ml$^{-1}$) at 319 nm.

The optimum $\Delta\lambda$ is important for the sensitivity of measuring in synchronous fluorimetry. TRH and ITR synchronous fluorescence spectra were recorded over different $\Delta\lambda$ (20–100 nm). It was concluded that both TRH and ITR could be measured optimally at $\Delta\lambda$ 60 nm with good smooth peaks. Decreasing $\Delta\lambda$ below this value exhibited lower fluorescence intensity for both compounds while $\Delta\lambda$ more than 60 nm caused overlapping spectra.

The pH impact was explored using 0.2 M acetate buffer (pH 3.6–5.6) and 0.2 M borate buffer (pH 7.0–10.0). Different pH values did not enhance the synchronous fluorescence intensity. Then, no buffer was used throughout the study, and this was also matched with the previous comparison methods [5,11].

The spectra of the two drugs were compared in different solvents. Water and alcohols—methanol and ethanol and $n$-propanol—were investigated; $n$-propanol resulted in high blank readings, while methanol and ethanol decreased the fluorescence intensity. Fortunately, the fluorescence intensities of TRH and ITR were better in distilled water compared to other solvents. Accordingly, distilled water was the ideal solvent resulting in the highest fluorescence intensities for TRH and ITR with a green impact on the environment.

To enhance the sensitivity of measuring different surfactants were tried—cetrimide, SDS, methyl cellulose and Tween 80—above their reported critical micelle concentration [27]. The mentioned surfactants did not increase or enhance the shape of the spectra of the two drugs.

## 3.2. Validation of the method

To consider the proposed method a validated procedure, the guidelines of International Conference on Harmonization (ICH) Q2 (R1) were followed [28]. The calculated data are summarized in table 1.

**Table 1.** Analytical performance data of the proposed first derivative synchronous spectrofluorimetric method.

| parameter | ITR | TRH |
|---|---|---|
| wavelength (nm) | 319 | 257 |
| linearity range (µg ml$^{-1}$) | 0.5–4.0 | 0.1–0.7 |
| intercept (*a*) | −3.031 | −2.76 |
| slope (*b*) | 12.041 | 80.40 |
| correlation coefficient (*r*) | 0.9997 | 0.9998 |
| s.d. of residuals ($S_{y/x}$) | 0.503 | 0.35 |
| s.d. of intercept ($S_a$) | 0.395 | 0.34 |
| s.d. of slope ($S_b$) | 0.154 | 0.76 |
| percentage relative standard deviation, % RSD | 1.45 | 1.34 |
| percentage relative error, % error | 0.59 | 0.54 |
| limit of detection, LOD (µg ml$^{-1}$) | 0.1085 | 0.014 |
| limit of quantitation, LOQ (µg ml$^{-1}$) | 0.329 | 0.042 |

It was found that linear correlation between peak amplitudes and concentration was provided over the range of 0.1–0.7 µg ml$^{-1}$ for TRH and 0.5–4.0 µg ml$^{-1}$ ITR at 257 and 319 nm, respectively. Limits of quantitation and limits of detection are calculated and abridged in table 1. Equations that represent linear regression analysis are

$$^1D = 80.40C - 2.67 \ (r = 0.9998) \ \text{for TRH at 257 nm}$$

$$^1D = 12.04C - 3.03 \ (r = 0.9997) \ \text{for ITR at 319 nm.}$$

Where $^1D$ is the peak amplitude, $C$ is the concentration of the drug in µg ml$^{-1}$ and $r$ is the correlation coefficient.

To test the accuracy of the proposed synchronous spectrofluorometric method, the obtained results were compared with previous native spectrofluorometric methods for both drugs [5,11]. The comparison method for TRH is a native spectrofluorimetric method that measured the drug at 336 nm after excitation at 275 nm [5]. The native fluorescence intensity of ITR is measured at 380 nm after excitation at 260 nm [11]. Additionally, statistical analysis of data indicated no significant difference between the methods as shown in table 2 [29].

The proposed method was also tested with regard to intra-day and inter-day precision (table 3). Three concentrations within the calibration curve for each drug were examined and the relative standard deviations were found to be small proving the repeatability and intermediate precision of the proposed method.

## 3.3. Applications

### 3.3.1. Analysis of TRH and ITR in their synthetic mixtures and pharmaceutical preparations

Different concentrations from the two studied drugs were determined in varied ratios in synthetic mixtures. Analysis of these mixtures shows the applicability of the designed method for their selective determination, as represented in table 4. Figure 6 demonstrates the good spectral resolution for TRH and ITR in their synthetic mixture. Furthermore, single ingredient tablets for each drug were analysed by the proposed method to test the specificity and the probability of interference from the excipients, maize starch, talc and lactose monohydrate. The results in table 5 reveal good percentage recoveries and standard deviations. Moreover, the comparison of the results with previously published ones get acceptable statistical data regarding Student's *t*-test and variance ratio *F*-test [29].

### 3.3.2. Analysis of TRH and ITR in spiked human plasma

Peak plasma levels of about 1.0 mg l$^{-1}$ of TRH occur 2 h after 250 mg single oral dose and the therapeutic concentration of ITR is reported to be greater than 0.25 mg l$^{-1}$ [26]. The sensitivity of the proposed

**Table 2.** Application of the proposed method to the determination of ITR and TRH in their pure form. Each result is the average of three separate determinations. The values between parentheses are the tabulated $t$ and $F$ values at $p = 0.05$ [29].

| compound | proposed method | | | comparison methods [5,11] | |
|---|---|---|---|---|---|
| | amount taken (µg ml$^{-1}$) | amount found (µg ml$^{-1}$) | % found | amount taken (µg ml$^{-1}$) | % found |
| ITR | 0.5 | 0.5009 | 100.20 | 0.20 | 98.50 |
| | 0.7 | 0.6877 | 98.29 | 0.40 | 101.25 |
| | 1.7 | 1.6802 | 98.82 | 0.80 | 99.75 |
| | 2.7 | 2.7515 | 101.93 | | |
| | 3.5 | 3.5322 | 100.91 | | |
| | 4.0 | 3.9475 | 98.68 | | |
| mean | | 99.81 | | | 99.83 |
| ± s.d. | | 1.44 | | | 1.38 |
| $t$ | | | 0.03 (2.36) | | |
| $F$ | | | 1.09 (19.3) | | |
| TRH | 0.10 | 0.1017 | 101.70 | 0.05 | 98.00 |
| | 0.30 | 0.2944 | 98.13 | 0.13 | 102.31 |
| | 0.35 | 0.3554 | 101.54 | 0.15 | 98.67 |
| | 0.45 | 0.4493 | 99.84 | | |
| | 0.55 | 0.5469 | 99.44 | | |
| | 0.70 | 0.7024 | 100.34 | | |
| mean | | 100.17 | | | 99.66 |
| ± s.d. | | 1.35 | | | 2.32 |
| $t$ | | | 0.35 (2.36) | | |
| $F$ | | | 2.95 (19.29) | | |

**Table 3.** Inter-day and intra-day precision of the developed method. Each result is the average of three separate determinations.

| drug | conc. (µg ml$^{-1}$) | intra-day | | | Inter-day | | |
|---|---|---|---|---|---|---|---|
| | | mean ± s.d | % RSD | % error | mean ± s.d. | % RSD | % error |
| ITR | 1.0 | 99.13 ± 1.74 | 1.75 | 1.01 | 100.02 ± 1.01 | 1.01 | 0.59 |
| | 2.0 | 99.67 ± 0.15 | 0.15 | 0.09 | 99.52 ± 0.82 | 0.82 | 0.47 |
| | 3.0 | 100.33 ± 0.35 | 0.35 | 0.2 | 99.93 ± 0.47 | 0.47 | 0.27 |
| TRH | 0.20 | 100.23 ± 0.59 | 0.59 | 0.34 | 99.09 ± 0.34 | 0.34 | 0.20 |
| | 0.4 | 100.7 ± 0.82 | 0.81 | 0.47 | 99.7 ± 0.46 | 0.46 | 0.27 |
| | 0.6 | 100.4 ± 0.4 | 0.40 | 0.23 | 100.6 ± 0.72 | 0.72 | 0.41 |

method was down to 0.042 and 0.329 for TRH and ITR, respectively; therefore, it was applicable to estimate the two drugs in biological level. Using the proposed trajectory, a linear relationship was constructed in plasma samples spiked with TRH and ITR by plotting the amplitudes of the first derivative spectra versus the drug concentration (table 6).

Linear regression analysis of the data gave the following equations:

$$^1D = 8.025 + 3.9C \ (r = 0.9979) \text{ for TRH}$$

$$^1D = 10.93 + 4.2C \ (r = 0.9983) \text{ for ITR.}$$

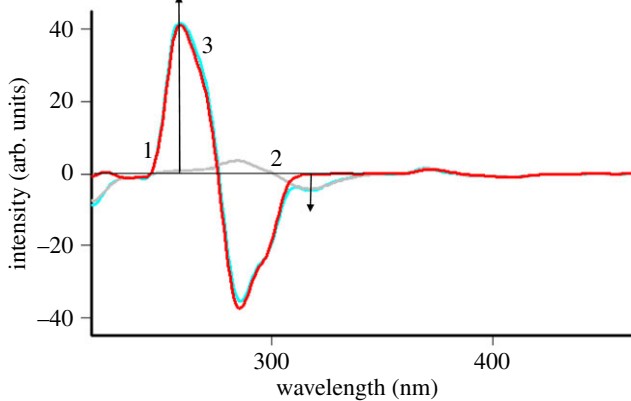

**Figure 6.** First derivative synchronous fluorescence spectra of: (1) 0.5 μg ml$^{-1}$ TRH, (2) 0.5 μg ml$^{-1}$ ITR, and (3) Mixture of 0.5 μg ml$^{-1}$ TRH and 0.5 μg ml$^{-1}$ ITR.

**Table 4.** Application of the proposed method to the determination of ITR and TRH in their synthetic mixtures.

| synthetic mixture | amount taken (μg ml$^{-1}$) | | % found | |
| --- | --- | --- | --- | --- |
| | ITR | TRH | ITR | TRH |
| | 0.6 | 0.6 | 98.67 | 100 |
| | 1.0 | 0.5 | 101.1 | 100.4 |
| | 2.0 | 0.5 | 99.85 | 99.60 |
| mean | | | 99.87 | 100 |
| ± s.d. | | | 1.22 | 0.4 |

**Table 5.** Application of the proposed method to the determination of ITR and TRH in their pharmaceutical preparations. Values between parentheses are the tabulated $t$ and $F$ values at $p = 0.05$ [29].

| compound | proposed method | | | comparison methods [5,11] | | |
| --- | --- | --- | --- | --- | --- | --- |
| | amount taken (μg ml$^{-1}$) | amount found (μg ml$^{-1}$) | % found | amount taken (μg ml$^{-1}$) | amount found (μg ml$^{-1}$) | % found |
| Itranox® tablet | 1.0 | 0.994 | 99.40 | 0.2 | 0.195 | 97.50 |
| | 2.0 | 2.013 | 100.65 | 0.4 | 0.408 | 102.00 |
| | 3.0 | 2.994 | 99.80 | 0.8 | 0.797 | 99.63 |
| mean | | | 99.95 | | | 99.71 |
| ± s.d. | | | 0.64 | | | 2.25 |
| $t$ | | | 0.17 (2.77) | | | |
| $F$ | | | 12.36 (19.00) | | | |
| lamisil® tablet | 0.2 | 0.198 | 99.00 | 0.05 | 0.050 | 100.00 |
| | 0.4 | 0.404 | 101 | 0.13 | 0.131 | 100.77 |
| | 0.6 | 0.598 | 99.67 | 0.15 | 0.149 | 99.33 |
| mean | | | 99.89 | | | 100.03 |
| ± s.d. | | | 1.02 | | | 0.72 |
| $t$ | | | 0.18 (2.77) | | | |
| $F$ | | | 2.01 (19.00) | | | |

**Table 6.** Assay results for the determination of the studied drugs in spiked human plasma samples using the proposed method.

| parameter | amount taken (µg ml$^{-1}$) | | amount found (µg ml$^{-1}$) | | % found | |
|---|---|---|---|---|---|---|
| | ITR | TRH | ITR | TRH | ITR | TRH |
| | 0.70 | 0.30 | 0.7071 | 0.3013 | 101.01 | 100.43 |
| | 0.80 | 0.40 | 0.7905 | 0.4038 | 98.81 | 100.95 |
| | 0.90 | 0.60 | 0.8976 | 0.5833 | 99.73 | 97.22 |
| | 1.00 | 0.70 | 1.0048 | 0.7115 | 100.48 | 101.64 |
| $x$ | | | | | 100.01 | 100.06 |
| ± s.d. | | | | | 0.96 | 1.96 |
| % RSD | | | | | 0.96 | 1.96 |
| % error | | | | | 0.48 | 0.98 |

# 4. Conclusion

The commonly prescribed drugs TRH and ITR are used for treatment and prophylaxis of systemic fungal infection. To accomplish the goal of their simultaneous determination, effortless and fast methodology was developed based on first derivative synchronous spectrofluorimetry. The method permits rapid measuring using water as an economic and safe diluting solvent. Analysis of synthetic mixtures and single ingredient tablets was carried out to prove selectivity of the method. As the two drugs are commonly prescribed together, the method was applicable to their assay in spiked human plasma using the optimized trajectory. The advantages of the proposed method include enhancing selectivity, resolution and low analysis time. This makes our proposed method an ideal one for the analysis of both drugs.

Ethics. Permission to collect samples was granted to El-Brashy *et al*. (code no. 2018-70). The study has been reviewed and reapproved by the ethical committee of Faculty of Pharmacy, Mansoura University (code no. 2020-21).
Data accessibility. Data are available from the Dryad Digital Repository: https://doi.org/10.5061/dryad.37pvmcvgc [30].
Authors' contributions. A.R. carried out the laboratory work, participated in data analysis and drafted the manuscript; H.E. and S.S. participated in the design of the study and carried out the statistical analyses; A.E.-B. designed the study, coordinated the study and revised the manuscript. All authors approved the manuscript for publication.
Competing interests. We have no competing interests.
Funding. No funding supported this research.

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
