## [Reviewer comments · Royal Society Open Science]

Review History

RSOS-200571.R0 (Original submission)

Review form: Reviewer 1

Is the manuscript scientifically sound in its present form?

Yes

Are the interpretations and conclusions justified by the results?

Yes

Is the language acceptable?

Yes

Do you have any ethical concerns with this paper?

No

Have you any concerns about statistical analyses in this paper?

No

Recommendation?

Accept with minor revision (please list in comments)

Comments to the Author(s)

Herein a simple and sensitive spectrofluorimetric method developed for simultaneous determination of terbinafine and itraconazole. The rationale for simultaneous measurement of the two antifungal drugs is based on potential combination of both drugs in the treatment of fungal infections.

The method is characterized by novelty, however the manuscript need revision, before recommendation for publication can be justified.

The points that need to be revised during revision are as follows:

-Page 3, line 8: The abbreviation (SFS) should be added after “synchronous fluorescence spectroscopy”.

-Page 3, from line 13 to 29, need to be re-written in a better way. The authors should clarify if there are any reports regarding simultaneous analysis of the two drugs. Authors might consider adding some more recent reviews, e.g., in AJPCR 2019, in IJPPR 2019.

-Page 4, line 39: (no reported methodology has been published for the concurrent determination of TRH and ITR). “It should be (no reported flourimetric method.....”

- Under the subtitle “Standard solutions”

- The authors should clarify that the two drugs were separately dissolved in methanol.

- Page 4, line 42: The authors mentioned that “Subsequent dilution of the standard solutions with methanol was carried out to get the working solutions”.

It was not clear if the subsequent dilution was with methanol or with methanol then with water.

-Under the subtitle “Construction of calibration graphs”

Rephrase the sentence to begin with “Samples of TRH and ITR solutions.....”

-Page 5, line 28: change the title to “Analysis of TRH and ITR in spiked human plasma”.

- Analysis of the studied drugs in spiked human plasma:

- Plasma samples filtration step before analysis was missed here.

- Why the authors decide to dilute 1 mL from the aspirated upper layers to 10 mL with water.

The use of lower volume of water for dilution is preferred to measure the content of these substances in real plasma samples.

-Page 6, line 42: “water and short chain alcohols were investigated”. The authors are advised to clarify which alcohols were used and the effect of each solvent on the RFI of the studied drugs.

-Page 5, line 51: (reported critical micelle concentration). A reference could be mentioned here.

-Under the subtitle “Analysis of TRH/ITR in their synthetic mixtures and pharmaceutical preparations”:

-Change “TRH/ITR to “TRH and ITR”.

-The authors should clarify if the analysis of the pharmaceutical preparations was in the native mode.

- Change “in presence of their synthetic mixture” to “in their synthetic mixture”.

- Under conclusion title:

Rephrase sentence so that it doesn't start with an abbreviation.

- Table 3: The title should be “Intra-day and inter-day precision of the developed method”

- Table 5 was overlapping table 4 and the results are not clear.

- Table 6: some of the % found recovery data need correction.

Review form: Reviewer 2

Is the manuscript scientifically sound in its present form?

Yes

Are the interpretations and conclusions justified by the results?

Yes

Is the language acceptable?

Yes

Do you have any ethical concerns with this paper?

No

Have you any concerns about statistical analyses in this paper?

No

Recommendation?

Accept with minor revision (please list in comments)

Comments to the Author(s)

This is overall a solid piece of work for the simultaneous quantification of the drugs Terbinafine and Itraconazole. My only comment is that the Tables 4 and 5 are overlapping in the present version of this manuscript. The authors should also consider to move some of the data in the tables into a separate file as an Electronic Appendix.

Decision letter (RSOS-200571.R0)

Dear Dr Elmansi:

Title: Combining derivative and synchronous approaches for simultaneous spectrofluorimetric determination of Terbinafine and Itraconazole.

Manuscript ID: RSOS-200571

Thank you for submitting the above manuscript to Royal Society Open Science. On behalf of the Editors and the Royal Society of Chemistry, I am pleased to inform you that your manuscript will be accepted for publication in Royal Society Open Science subject to minor revision in accordance with the referee suggestions. Please find the reviewers' comments at the end of this email.

The reviewers and handling editors have recommended publication, but also suggest some minor revisions to your manuscript. Therefore, I invite you to respond to the comments and revise your manuscript.

Because the schedule for publication is very tight, it is a condition of publication that you submit the revised version of your manuscript before 01-Jul-2020. Please note that the revision deadline will expire at 00.00am on this date. If you do not think you will be able to meet this date please let me know immediately.

When submitting your revised manuscript, you will be able to respond to the comments made by the referees and upload a file "Response to Referees" in "Section 6 - File Upload". You can use this to document any changes you make to the original manuscript. In order to expedite the

processing of the revised manuscript, please be as specific as possible in your response to the referees.

Kind regards,
Dr Laura Smith
Publishing Editor, Journals

On behalf of the Subject Editor Professor Anthony Stace and the Associate Editor Dr Debashree Ghosh.

RSC Associate Editor:
Comments to the Author:
(There are no comments.)

RSC Subject Editor:
Comments to the Author:
(There are no comments.)

Reviewer comments to Author:

Reviewer: 1

Comments to the Author(s)

Herein a simple and sensitive spectrofluorimetric method developed for simultaneous determination of terbinafine and itraconazole. The rationale for simultaneous measurement of the two antifungal drugs is based on potential combination of both drugs in the treatment of fungal infections.

The method is characterized by novelty, however the manuscript need revision, before recommendation for publication can be justified.

The points that need to be revised during revision are as follows:

-Page 3, line 8: The abbreviation (SFS) should be added after "synchronous fluorescence spectroscopy".

-Page 3, from line 13 to 29, need to be re-written in a better way. The authors should clarify if there are any reports regarding simultaneous analysis of the two drugs. Authors might consider adding some more recent reviews, e.g., in AJPCR 2019, in IJPPR 2019.

-Page 4, line 39: (no reported methodology has been published for the concurrent determination of TRH and ITR). "It should be (no reported flourimetric method....."

- Under the subtitle "Standard solutions"

- The authors should clarify that the two drugs were separately dissolved in methanol.

- Page 4, line 42: The authors mentioned that "Subsequent dilution of the standard solutions with methanol was carried out to get the working solutions".

It was not clear if the subsequent dilution was with methanol or with methanol then with water.

-Under the subtitle "Construction of calibration graphs"

Rephrase the sentence to begin with "Samples of TRH and ITR solutions....."

-Page 5, line 28: change the title to "Analysis of TRH and ITR in spiked human plasma".

- Analysis of the studied drugs in spiked human plasma:

- Plasma samples filtration step before analysis was missed here.

- Why the authors decide to dilute 1 mL from the aspirated upper layers to 10 mL with water.

The use of lower volume of water for dilution is preferred to measure the content of these substances in real plasma samples.

-Page 6, line 42: "water and short chain alcohols were investigated". The authors are advised to clarify which alcohols were used and the effect of each solvent on the RFI of the studied drugs.

-Page 5, line 51: (reported critical micelle concentration). A reference could be mentioned here.

-Under the subtitle" Analysis of TRH/ITR in their synthetic mixtures and pharmaceutical preparations":

-Change "TRH/ITR to "TRH and ITR".

-The authors should clarify if the analysis of the pharmaceutical preparations was in the native mode.

- Change "in presence of their synthetic mixture" to "in their synthetic mixture".

- Under conclusion title:

Rephrase sentence so that it doesn't start with an abbreviation.

- Table 3: The title should be "Intra-day and inter-day precision of the developed method"

- Table 5 was overlapping table 4 and the results are not clear.

- Table 6: some of the % found recovery data need correction.

Reviewer: 2

Comments to the Author(s)

This is overall a solid piece of work for the simultaneous quantification of the drugs Terbinafine and Itraconazole. My only comment is that the Tables 4 and 5 are overlapping in the present version of this manuscript. The authors should also consider to move some of the data in the tables into a separate file as an Electronic Appendix.

Author's Response to Decision Letter for (RSOS-200571.R0)

See Appendix A.

Decision letter (RSOS-200571.R1)

Dear Dr Elmansi:

Title: Combining derivative and synchronous approaches for simultaneous spectrofluorimetric determination of Terbinafine and Itraconazole.

Manuscript ID: RSOS-200571.R1

It is a pleasure to accept your manuscript in its current form for publication in Royal Society Open Science. The chemistry content of Royal Society Open Science is published in collaboration with the Royal Society of Chemistry.

On behalf of the Subject Editor Professor Anthony Stace and the Associate Editor Dr Debashree Ghosh.

RSC Associate Editor
Comments to the Author:
(There are no comments.)

Reviewer(s)' Comments to Author:

Appendix A

On behalf of my co-authors, I wanted to thank the editorial team and reviewers for accepting our manuscript to be published, regarding the help, assistance and comments. We would like to let you know that we appreciate the time you took to revise our work. All the comments were considered and we did our best to modify and correct all the comments:

Reviewer 1:

- Page 3, line 8: The abbreviation (SFS) should be added after “synchronous fluorescence spectroscopy”.

Reply: The abbreviation was added as recommended by the reviewer.

- Page 3, from line 13 to 29, need to be re-written in a better way. The authors should clarify if there are any reports regarding simultaneous analysis of the two drugs. Authors might consider adding some more recent reviews, e.g., in AJPCR 2019, in IJPPR 2019.

Reply: This section was re-written according to the reviewer comment. The two mentioned reviews were included in references and cited in the manuscript [25 ,26].

- Page 4, line 39: (no reported methodology has been published for the concurrent determination of TRH and ITR.). “It should be (no reported fluorometric method.....”

Reply: The paragraph including this sentence was re-written in a better way.

- Under the subtitle “Standard solutions”
- The authors should clarify that the two drugs were separately dissolved in methanol.

Reply: The required modification has been carried out as recommended.

- Page 4, line 42: The authors mentioned that “Subsequent dilution of the standard solutions with methanol was carried out to get the working solutions”. It was not clear if the subsequent dilution was with methanol or with methanol then with water.

Reply: the subsequent dilution to obtain the working solutions was with methanol. After that, aliquots from the working solutions were taken into 10 mL volumetric flasks and completed to volume by distilled water. So water is the diluting solvent in the main procedure.

- Rephrase the sentence to begin with “Samples of TRH and ITR solutions.....” Page 5, line 28: change the title to “Analysis of TRH and ITR in spiked human plasma”. Plasma samples filtration step before analysis was missed here.

Reply: The required changes have been performed and the filtration step was added.

- Why the authors decide to dilute 1 mL from the aspirated upper layers to 10 mL with water. The use of lower volume of water for dilution is preferred to measure the content of these substances in real plasma samples.

Reply: The volumes were diluted to be completed in 10-mL volumetric flasks to proceed as in the construction of calibration curves. As we are dealing with spiked human plasma, the dilution

wasn't expected to make an error. But if it is real plasma sample, the reviewer comment is very important as the dilution shouldn't reach 1:10 ratio.

- Page 6, line 42: "water and short chain alcohols were investigated". The authors are advised to clarify which alcohols were used and the effect of each solvent on the RFI of the studied drugs.

Reply: the required clarification has been performed as recommended by the reviewer.

- Page 5, line 51: (reported critical micelle concentration). A reference could be mentioned here.

Reply: A reference has been added as required with reference No. [27].

- Under the subtitle" Analysis of TRH/ITR in their synthetic mixtures and pharmaceutical preparations":
Change "TRH/ITR to "TRH and ITR".

Reply: the change has been performed as recommended.

- The authors should clarify if the analysis of the pharmaceutical preparations was in the native mode.
- Change "in presence of their synthetic mixture" to "in their synthetic mixture".

Reply: All the applications were carried out in the first derivative synchronous mode. This was clarified and the required correction in the title has been carried out.

- Under conclusion title:

Rephrase sentence so that it doesn't start with an abbreviation.

Reply: The required modification has been made as suggested by the reviewer.

- Table 3: The title should be "Intra-day and inter-day precision of the developed method"

Reply: the title has been edited as recommended.

- Table 5 was overlapping table 4 and the results are not clear.

Reply: I noticed this problem during the PDF construction, and I didn't know the source of the problem so I attached a separate PDF for the whole manuscript. The word file is completely replaced now, and I hope this problem now is solved.

- Table 6: some of the % found recovery data need correction.

Reply: the data were revised and corrected as recommended by the reviewer.

Reviewer: 2

- This is overall a solid piece of work for the simultaneous quantification of the drugs Terbinafine and Itraconazole. My only comment is that the Tables 4 and 5 are overlapping in the present version of this manuscript. The authors should also consider to move some of the data in the tables into a separate file as an Electronic Appendix.

Reply: all the tables was revised and corrected. We tried to put only the tables that are directly related to the manuscript findings. However, if the reviewer feels that certain table could be transferred to the electronic appendix we could make it.